# The evaluation of a tissue-engineered cardiac patch seeded with hips derived cardiac progenitor cells in a rat left ventricular model

Yuichi Matsuzaki[1][☯], Shinka Miyamoto[1][☯], Hideki Miyachi[1], Tadahisa Sugiura[1], James W. Reinhardt[1], Chang Yu-Chun[1], Jacob Zbinden[1], Christopher K. Breuer[1,2], Toshiharu Shinoka[1,3]*

1 Center for Regenerative Medicine, The Abigail Wexner Research Institute at Nationwide Children's Hospital, Columbus, OH, United States of America, 2 Department of Surgery, Nationwide Children's Hospital, Columbus, OH, United States of America, 3 Department of Cardiothoracic Surgery, The Heart Center, Nationwide Children's Hospital, Columbus, OH, United States of America

☯ These authors contributed equally to this work.
* toshiharu.shinoka@nationwidechildrens.org

## Abstract

### Background

Ventricular septal perforation and left ventricular aneurysm are examples of potentially fatal complications of myocardial infarction. While various artificial materials are used in the repair of these issues, the possibility of associated infection and calcification is non-negligible. Cell-seeded biodegradable tissue-engineered patches may be a potential solution. This study evaluated the feasibility of a new left ventricular patch rat model to study neotissue formation in biodegradable cardiac patches.

### Methods

Human induced pluripotent stem cell-derived cardiac progenitor cells (hiPS-CPCs) were cultured onto biodegradable patches composed of polyglycolic acid and a 50:50 poly (l-lactide-co-ε-caprolactone) copolymer for one week. After culturing, patches were implanted into left ventricular walls of male athymic rats. Unseeded controls were also used (n = 10/group). Heart conditions were followed by echocardiography and patches were subsequently explanted at 1, 2, 6, and 9 months post-implantation for histological evaluation.

### Result

Throughout the study, no patches ruptured demonstrating the ability to withstand the high pressure left ventricular system. One month after transplantation, the seeded patch did not stain positive for human nuclei. However, many new blood vessels formed within patches with significantly greater vessels in the seeded group at the 6 month time point. Echocardiography showed no significant difference in left ventricular contraction rate between the two groups. Calcification was found inside patches after 6 months, but there was no significant difference between groups.

**Data Availability Statement:** All relevant data are within the paper.

**Funding:** This research is supported by US National Institutes of Health (NIH) grants: R01HL098228 and GUNZE co ltd. Specifically GUNZE co ltd provided material support through the creation of the biodegradable patch. YM was supported by Department of Defence (DoD) and Funding award from Uehara Memorial Foundation (Tokyo Japan) in 2019. JWR was supported in part by the American Heart Association under Award Number 18POST33990231. The specific roles of these authors are articulated in the 'author contributions' section.

**Competing interests:** The authors have read the journal's policy and the authors of the manuscript have the following competing interests: CKB and TS have received grant support from Gunze Ltd. No authors have received salary support from Gunze Ltd. This does not alter our adherence to PLOS ONEpolicies on sharing data and materials. There are no patents, products in development, or marketed products to declare.

## Conclusion

We have developed a surgical method to implant a bioabsorbable scaffold into the left ventricular environment of rats with a high survival rate. Seeded hiPS-CPCs did not differentiate into cardiomyocytes, but the greater number of new blood vessels in seeded patches suggests the presence of cell seeding early in the remodeling process might provide a prolonged effect on neotissue formation. This experiment will contribute to the development of a treatment model for left ventricular failure using iPS cells in the future.

## Introduction

Due to the complication of myocardial infarction in adults, problems such as left ventricular rupture and ventricular septal perforation may present as fatal complications, mandating surgical repair. [1] The materials currently utilized are not degradable, lack growth potential, and do not promote attachment of endotherial and smooth muscle cells or macrophage migration. [2] Tissue engineering may provide a new way to regenerate organs and deliver tissue-specific cell types to the site of injury, thereby replicating the structure and function of native myocardium. [3] Tissue engineered constructs often consist of a scaffold and seeded cells; as the scaffold degrades, new biocompatible tissue is deposited. [4] Human iPS (hiPS) cells are an infinite source of cardiomyocytes due to their high potential for differentiation and are therefore reported to be one of the most promising cell sources for cardiac regeneration therapy. [5,6,7,8,9] Previously, we reported the development of tissue engineering technology using induced, or artificial, pluripotent stem (iPS) cells and demonstrated that this method is feasible and safe. [10,11] Limitations of these early studies include the use of differentiated cardiomyocytes that may as a result have limited mitotic potential and use of a right ventricular model. [11] We created a new cardiac patch in which hiPS cell-derived cardiac progenitor cells (hiPS-CPC) are seeded on a sheet of bioabsorbable polymer and implanted in a rat left ventricular myocardial defect model.The patch was implanted in the apex wall of the left ventricle so that the luminal surface of the graft received direct blood supply and allowed host cardiomyocyte ingrowth at the anastomosis. The purpose of this study was to verify whether it is possible to obtain improved neoplasia by seeding progenitor cells and to verify whether the material can compensate for left ventricular injury.

## Methods

### Preparation of the tissue-engineered cardiac patch

A scaffold composed of a woven fabric of polyglycolic acid (PGA) and 50:50 poly (l-lactic-co-ε-caprolactone) copolymer (PLCL) was constructed as previously described. [12] The scaffold with a thickness of 0.6 to 0.7 mm was cut into a 6-mm circular patch using a biopsy punch (Fig 1A). The structure of the sponge was checked by imaging using scanning electron microscopy (SEM, Fig 1B and 1C). hiPS-CPCs (Cellular Dynamics International (CDI), Madison, Wisconsin) were cultured on the biodegradable patch in a 96-well plate at a density of $1.0 \times 10^6$ cells per well for one week and cell culture media was changed every two days. Because these cells were myocardial progenitor cells, they did not beat before transplantation. After culturing, patches were implanted in the left ventricular wall of male nude athymic rats, as described below.

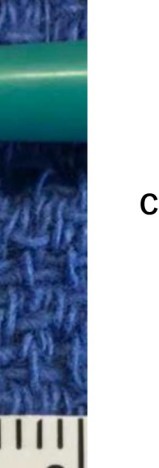

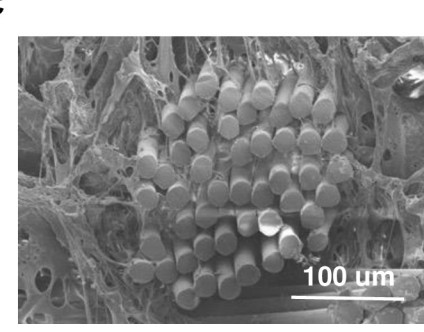

**Fig 1.** *Example of cardiac patch composed of knit mesh of PGA between 2 layers of PLCL*; A: 6-mm diameter biodegradable patch, B: SEM images of the sponge layer 100X magnification. C: SEM images of the sponge layer 500X magnification. This patch consists of a knit mesh of PGA sandwiched between 2 layers of PLCL.

## Left ventricle wall biodegradable patch implant procedure

This study was carried out in strict accordance with the recommendations in the Guide for the Care and Use of Laboratory Animals of the National Institutes of Health. The protocol was approved by the Committee on the Ethics of Animal Experiments of the Nationwide Children's Hospital (Protocol Number: AR12-00079). All surgery was performed under anesthesia and all efforts were made to minimize suffering. Adult male nude athymic rats (Jackson Laboratories, Bar Harbor, ME) weighing 230–300 g were used for the left ventricle wall replacement procedure. General anesthesia was induced with ketamine (50 mg/kg, i.p.) and xylazine (5 mg/kg, i.p.); anesthesia was maintained using isoflurane (1.5%) in oxygen. Animals were intubated with a 16-gauge catheter and respiration was maintained at 60 cycles per minute with a tidal volume of 2.5 ml. The surgical procedure was performed using aseptic techniques with sterile instruments. First, the skin of the chest was sterilized with a povidone-iodine solution and the heart was exposed through a median sternotomy. A purse string suture was placed in the left ventricle apex wall with 7–0 polypropylene sutures (Ethicon, Somerville, NJ, USA). Both ends of the suture were passed through a 22-gauge plastic vascular cannula, which was used as a tourniquet (Fig 2A and 2B). The tourniquet was tightened and two-thirds of the distended part of the left ventricle apex wall inside the purse string suture was cut to create a flap. To indicate that a transluminal defect had been created in the left ventricle, the tourniquet was briefly released to determine whether massive bleeding occurred and a core of a 16-gauge catheter was inserted in left ventricle (Fig 2C–2E). Next, the cardiac patch was sutured along the margin of the purse string suture with 7–0 polypropylene to cover the hole in the left ventricle

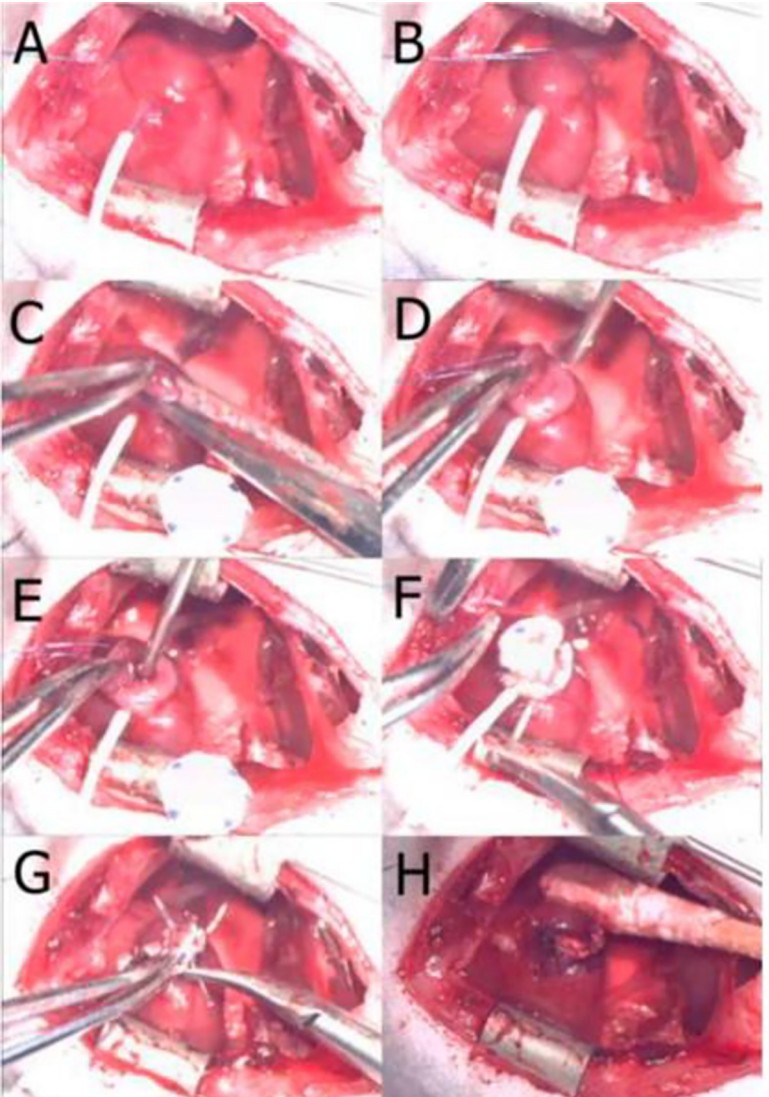

**Fig 2. Various stages of patch implantation.** (A): A purse string suture placement in the left ventricle apex wall; (B): tourniquet tightening; (C): cutting the distended part of the left ventricle apex wall; (D) flap creation; (E): needle used to examine hole creation; (F): cardiac patch implantation; (G): closing the flap (H): purse string suture removal.

(Fig 2F) and the ventricular flap was then sutured onto the patch (Fig 2G). After suturing, the tourniquet was released and the purse string suture was removed. Additional sutures were added, as needed, to achieve hemostasis (Fig 2H). After expanding the lungs using positive end-expiratory pressure, the sternum was closed parasternally with four interrupted 5–0 poly-propylene sutures (Ethicon). The muscle layer and skin were closed with 4–0 Vicryl absorbable sutures (Ethicon). For the first 3 days after surgery, buprenorphine (0.05 mg/kg) and cefurox-ime (100 mg/kg) were administered subcutaneously twice daily. Animals were randomly divided into two groups: 1) hiPS-CPC seeded group and 2) unseeded group (n = 10/group). Two animals in each group were sacrificed at 1 and 2 months post-surgery and 3 animals in each group were sacrificed at 6 and 9 months post-surgery.

## Echocardiographic analysis

Echocardiographic measurements were obtained preoperatively and at 1, 2, 6, and 9 months postoperatively. Animals were anesthetized with isoflurane (2% isoflurane with 100% oxygen gas inhalation through a nose cone). Echocardiography was performed to assess ventricular function (Vevo Visual Sonics 770; Visual Sonics, Toronto, ON, Canada). Two-dimensional imaging in a parasternal long axis plane was obtained. Left ventricular systolic and diastolic volumes were estimated by the modified Simpsons method to calculate LV ejection fraction (Analysis package from Vevo Visual Sonics).

## Histology, immunohistochemistry, and immunofluorescence

After sacrifice, the left ventricle was separated from the heart by axial dissection, and a sagittal transection centered on the apical patch was performed. Explanted cardiac patches were fixed in 4% paraformaldehyde, embedded in paraffin, and sectioned at a thickness of 5 μm. Sections were stained with hematoxylin and eosin (H&E), Masson's trichrome, von Kossa to examine calcification, and by immunohistochemistry and immunofluorescence (IF). The degree of calcification was determined by area measurements using ImageJ software. Measurements are reported as a percentage of positive area over the entire patch area.

For immunohistochemistry, tissue sections were blocked for endogenous peroxidase activity prior to staining. Platelet endothelial cell adhesion molecule (CD31, 1:1000, Abcam, Cambridge, MA) was used to identify endothelial cells. Antibody binding was detected with appropriate biotinylated secondary antibodies followed by streptavidin-horseradish peroxidase. Positive staining was assessed using 3,3-diaminobenzidine peroxidase substrate (Vector Laboratories SK4105, Burlingame, CA). Nuclei were counterstained with Gill's hematoxylin. Angiogenesis measurements reported as count of positive CD31 vascular over the entire patch area.

## Statistical analysis

Numeric values are listed as mean ± standard deviation (SD). Statistical analyses of ejection fractions were conducted via two-way ANOVA; all other data were analyzed via Student's t-test. A $p$ value less than 0.05 was selected as the cutoff for statistical significance.

# Results

## Postoperative course and gross observations

No deaths occurred during the postoperative course in either group and no gross evidence of thrombosis was present in any of the animal explants. At the time of explantation, all rat hearts exhibited minimal thoracic adhesions with no recognizable pattern of adhesive tissue present in either groups. There was no evidence of aneurysm formation at the site of the implanted patch in the left ventricle at any time point (Fig 3).

## Echocardiographic analysis

At all-time points, there was no significant difference in ejection fraction between the two groups (Fig 4). Diastolic volumes increased over time in both groups, but there was no significant difference. In the seeded group, there was a trend towards smaller ventricular systolic volumes, but this difference was also not statistically significant.

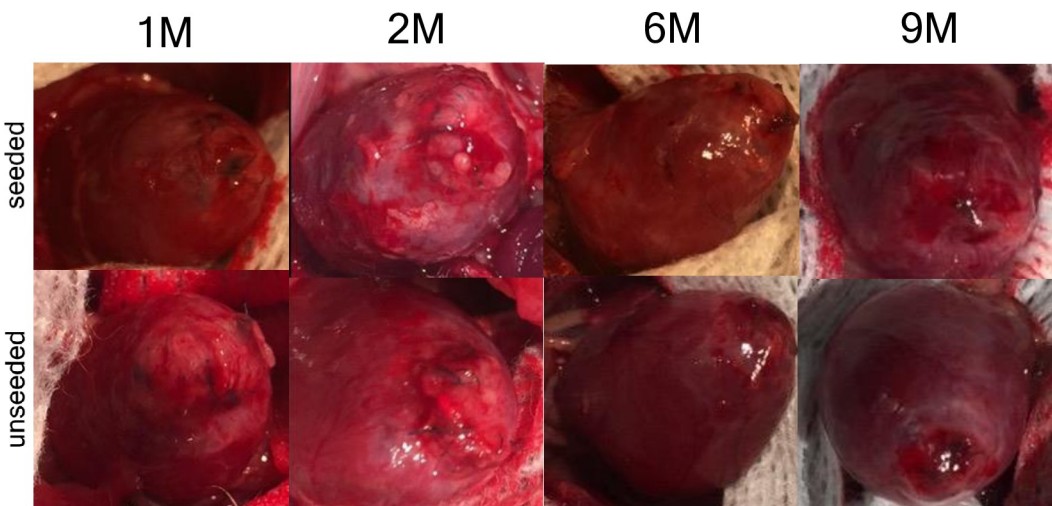

**Fig 3. Macroscopic view of operated rat hearts.** Over time, some fibrous adhesions on the pericardial surface formed. The patch was sutured in the heart interior. There were no significant differences in macroscopic findings between the two groups.

## Histology, immunohistochemistry, and immunofluorescence

Visualization of Masson's trichrome stain show collagen and muscle fibers. At eight weeks, some patch material still remains and is covered by collagen. Over the course of 24 weeks, the cardiac patch gradually degraded and could not be detected (Fig 5A). We assessed PGA fiber degradation and collagen formation using polarized light microscopy with Picro-Sirius Red (PSR) stained histological sections from 1, 2, 6 and 9 months after implantation. Under

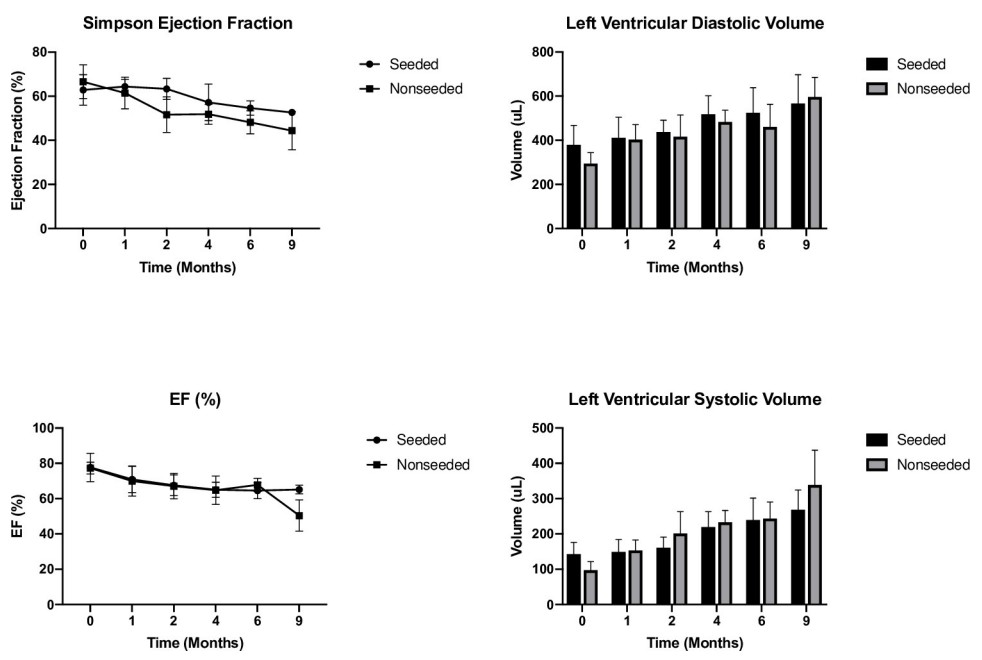

**Fig 4. Echographic analyses over a period of 36 weeks compared to baseline: LV Ejection fraction (EF); LV diastolic volume; LV systolic volume.** There are no aneurysmal changes in either group.

A

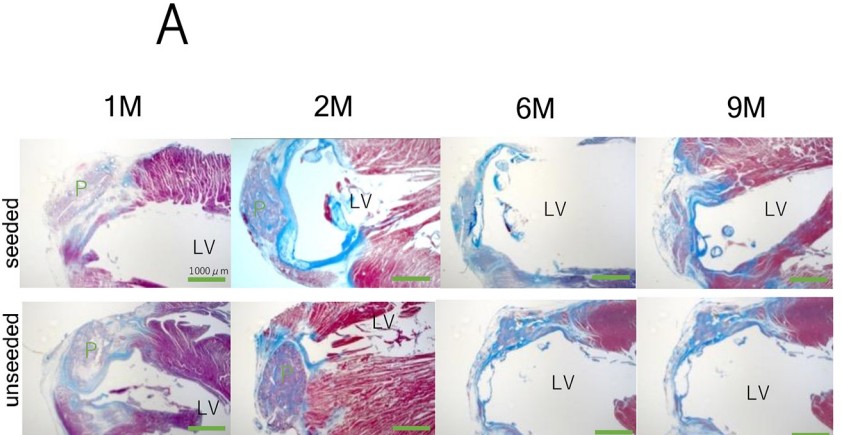

P : Patch, LV : Left Ventricle

B

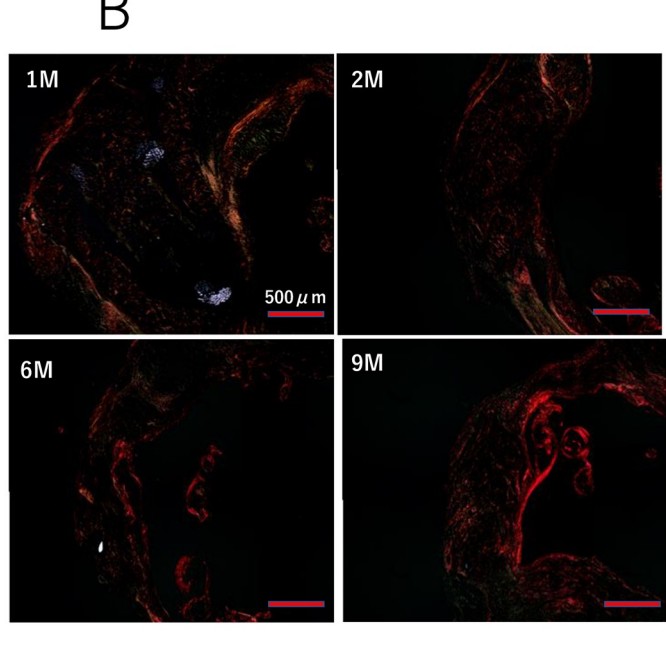

C

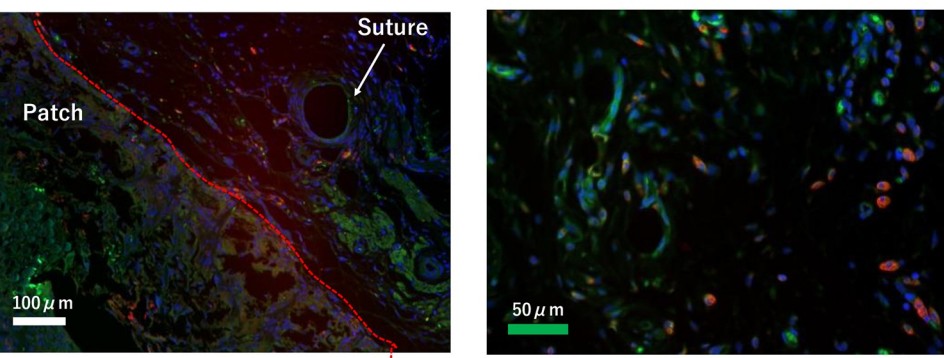

Red; Human nuclear antigen, Green: Troponin T   Blue: DAPI

**Fig 5.** (A) Masson's trichrome staining of sections indicating collagen (blue) and muscle (red) demonstrating cell infiltration and material degradation. (B) PSR staining of section indicating inmature collagen (yellow) and mature collagen (red), White fiber (PGA fiber) demonstrating PGA degraded at the 2-month time point (red bar 500μm), (C) Evaluation of iPS cells and cardiomyocytes on patches one month after transplantation using immunofluorescence. Red; human nuclear antigen, Green: troponin T Blue: DAPI.

polarized light, the PGA fibers are birefringent and can be visualized. The PGA fibers can be seen as highly organized bundles within the scaffold at 1M after implantation, but by 2M the fibers had fragmented and nearly degraded. No fragments of the PGA fibers were visible at 6 months after implantation (Fig 5B). The wall thickness of the LV ventricle apex was very thin. Using immunofluorescence, explanted patches were stained with anti-human nuclear antigen to detect hiPS-CM and troponin to detect cardiomyocytes. While images did reveal cells that stained positive for anti-human nuclear antigen, the pattern of staining was cytoplasmic, and not restricted to the nucleus as would be expected for this antigen. We therefore concluded that these cells represented false positives and remaining true hiPS-CM were not observed. (Fig 5C). At this time point, H&E staining showed cell infiltration. The number of cells present in the patch at each time point was compared, and there was no significant difference between the seeded group and the non-seeded group (Fig 6C-1). At 6 months, multinuclear giant cells and granular tissue can be observed indicating ongoing chronic inflammation (Fig 6A). After 6 months, von Kossa staining revealed no calcification in the patch (Fig 6B), but there was calcification present on the inner surface of the left ventricle. There was no significant difference in degree of calcification between two groups (Fig 6C-2), but there was a significant difference in the number of CD31 positive vessels (p = 0.03, Fig 6C-3).

## Discussion

Clinically, non-biodegradable materials are utilized as patches for left ventricular repair associated with sequelae after myocardial infarction; however, there are a host of problems related to durability, thrombosis, biocompatibility, and infection that requires reoperation. [2,12] For example, PTFE patches have been accepted worldwide as one of the most reliable non-biodegradable synthetic materials in terms of durability, low thrombotic properties, and comfortable handling during surgery, but there are many reports of bacterial infections. In general, cardiac surgeons have concerns about implanting foreign materials in the heart as infective endocarditis can be a lethal complication after open heart surgery. Even now, some surgeons inevitably use Dacron patch or PTFE patch to repair for ventricular septal rupture after myocardial infarction. In addition to these risks, left ventricular function after myocardial infarction continues to deteriorate following the use of non-degradable myocardial patches. These clinical demands motivated us to develop new bioabsorbable materials for use in cardiac surgery. Complications associated with non-biodegradable materials and the potential to regenerate functional myocardium associated with tissue engineering have fostered the development of surgical models and evaluation of various scaffold designs. Due to challenges and limitations in the use of exogenous cell-based biomaterials, cell-free strategies are being investigated as an alternative. [13] This approach requires the attachment and growth of host parenchymal cells as well as their production and organization of extracellular matrix. Materials of natural origin, such as collagen and fibrin, have been evaluated as scaffolds for tissue regeneration after myocardial infarction. These are applied surgically as patches, or less invasively as gel forming injectable ECM proteins. Acellular scaffolds are advantageous over cellular scaffolds in that: (1) these are off-the-shelf products that can be immediately implanted (e.g., SynerGraft®), (2) elicit a limited immune response, and (3) cost far less to produce. [14,15] Biodegradable synthetics, like the polymers that comprise our scaffold, are another category or materials widely

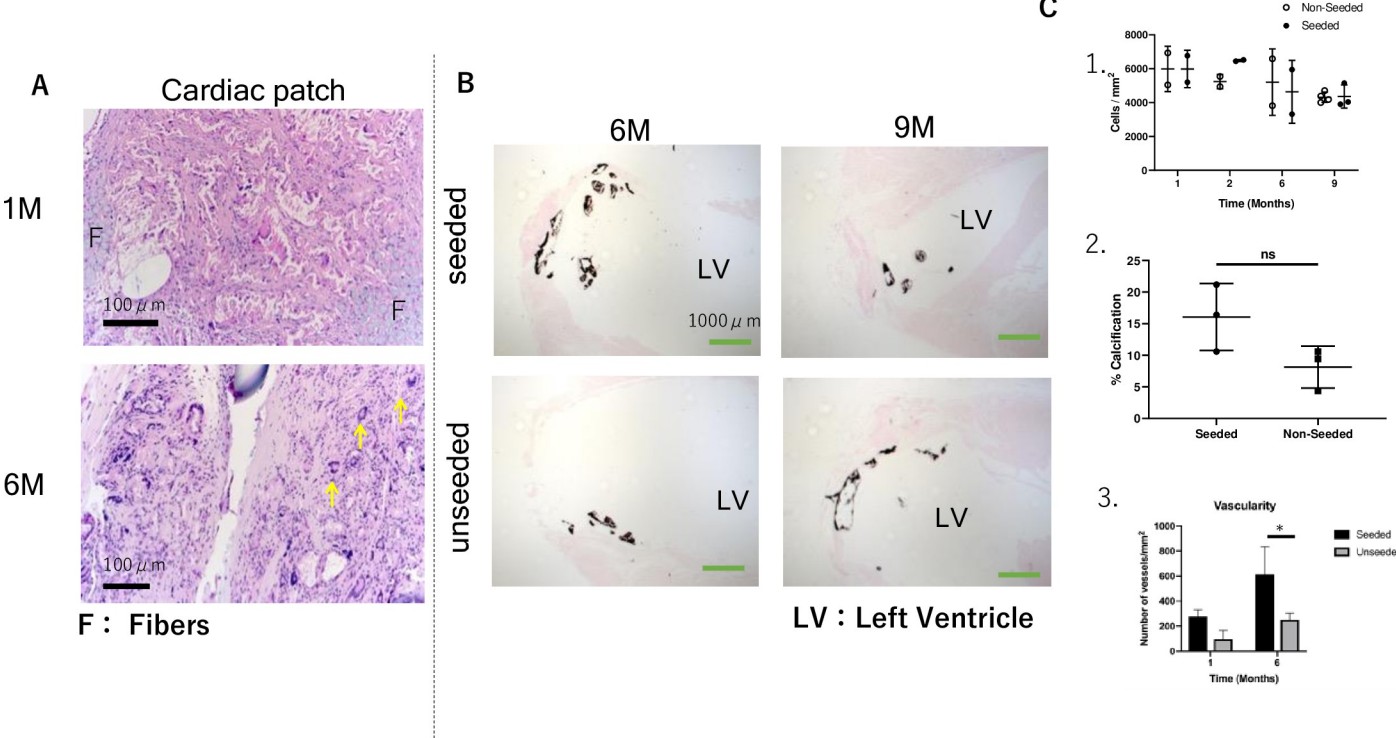

**Fig 6.** (A): Cells can infiltrate and attach to the scaffold four weeks after implantation. Hematoxylin and eosin staining revealed multinuclear giant cells (yellow arrows). (B): Von Kossa staining shows phosphate commonly associated with calcification in the LV at 24 and 36 weeks after implantation. (C):1. Number of cells in bioabsorbable patch at each time point. 2. Evaluation of calcification at 6 months after transplantation (n = 3). 3. Evaluation of the number of CD31 positive vessels 1 month and 6 months after implantation (n = 3/time point).

used in tissue engineering applications due to their mechanical properties, material uniformity, stability, and lower risk of infection compared to natural biomaterials. Biodegradable synthetic polymers can be modified with high precision to meet tissue-specific properties such as appropriate degradation rates, porosity, and mechanical strength. [16–19] Cell seeding can provide additional benefits to neotissue formation that include increasing functional capacity of resulting neotissue, accelerating neo tissue, formation, and modulating the immune response. However, it is well known in cell culture studies that cells are more likely to survive if they are able to attach to a surface as demonstrated in our experiments. [20] Proper selection of both a scaffold and seeded cells will continue to be an important determinant in the success of regenerative medicine. [4]

In the past, heart tissue was considered non-renewable; however, recent studies have suggested that cardiomyocytes, although rare, can replicate. In previous studies, Nakane et al. A designed heart tissue composed of three different cell types differentiated from hiPS cardiomyocytes (CM), endothelial cells, and vascular wall cells. [21] Compared to sham surgery, these structures have led to better preservation of viable myocardium and maintenance of wall thickness. This approach focuses on the outside of the heart, but the same methods applied inside the heart may be worth investigating.

This study focused on examining the feasibility of a biodegradable, prosthetic patch seeded with hiPS cells in improving left ventricular function following cardiac insult.

This study investigated the possibility of using a biodegradable PGA-PLCL scaffold for surgical left ventricular repair. Our previous work has shown proven that PGA-PLCL, an FDA-approved material with growth potential, seeded with cultured bone marrow cells used in the

venous system is clinically feasible. [22,23] This patch consists of a knit mesh of PGA sandwiched between 2 layers of PLCL. The PGA provides a strong backbone to withstand the blood flow within the body, while PLCL provides a porous surface for cell attachment and infiltration. Given that sufficient nutrient and oxygen delivery through functional vasculature or other means of perfusion is essential for successful cardiac stem cell therapy, we implanted the hiPS-CPC seeded heart patch via left ventricular reconstruction allowing the blood supply to be integrated directly into the luminal surface of the patch. Another consideration of the study was the thickness of graft material. A previous study by J. Riegler *et al*. demonstrated that the survival rate of seeded cells was lower when implanted in heart tissue grafts thicker than 400μm in mouse model. [24] Considering the difference in animal models, we chose a graft thickness of 600μm.

When we try to create infarction model with the additional incision of left ventricular apex for placing the patch in rats, animal survival rates were low following the procedure. Therefore, we simplified the model to utilize left ventricular apex incision to implant our patches in the blood stream. Following implantation, the patch was able to withstand the high-pressure system without rupturing and there was no evidence of infection in both groups. There is no report of survival experiments in rats which simulating the left ventricle, that is, simulating VSP or left ventricular rupture, and we consider the importance of this research on the method of surviving the rat for one year with no complication. However, long-term postoperative follow-up revealed calcification. These results suggest that cell seeding may not influence formation of calcification. Furthermore, as our previous works with PGA-PLCL tissue engineered constructs have exhibited no calcification in the venous and right ventricular environments, the results suggest that the cause may be due to shear stress of the left ventricle, extensive tissue necrosis, and the inflammatory response that can lead to to calcium chemo-attaction. [25] Immunofluorescence showed that seeded hiPS-CPCs do not remain on the biodegradable patch after 1 month. Prior *in vitro* studies on co-cultures of adult rat muscle and cardiac stem cells show that stem cells survive longer than adult cardiomyocytes alone due to increased levels of insulin-like growth factor 1 and vascular endothelial growth factors, both of which were associated with decreased apoptosis and increased myocyte survival. [26] The seeded cells may also contribute to differences in cardiac function. Keller et al. showed that scaffold-based 3D human dermal fibroblast cultures can be used as cardiac patches to stimulate revascularization when transplanted into infarcted LV of severe combined immunodeficiency (SCID) mice. [27] In our results, the cell seeding group also showed more revascularization at 6-month timepoints, however there was no statistical difference in left ventricular ejection fraction between the two groups during the 9-month follow-up.

Several limitations exist with this current work. Small sample size is one of the limitations. This study is still at a preliminary stage, and only examines how seeding iPS cells with bioabsorbable material contributes to myocardial regeneration. In the future study, we would like to perform power analysis and increase the number of samples. Our approach utilized cells purchased from CDI for which the genetic information and differentiation induction method are not clear; nor were the cells. [28] Therefore, we may need to generate a line of myocardial progenitor cells with our desired functions. Secondly, additional time-points are required to elucidate the temporal loss of seeded cells. It is possible that the in vivo structural and chemical environment results in the death of hiPS-CPCs.

Although more effort is required to further develop these cardiac patches, we have demonstrated the potential of tissue engineering to improve cardiac function following myocardial infarction.

## Conclusion

In summary, we have demonstrated the safety of a biodegradable cardiac patch of hiPS-CPC seeded in left ventricular myocardial contraction in rats. Seeded hiPS-CPCs disappeared from the patch at an early stage and did not contribute to left ventricular contractility, but may affect angiogenesis. These findings need to be further investigated for improved surgical outcomes.

## Author Contributions

**Conceptualization:** Yuichi Matsuzaki, Shinka Miyamoto, Tadahisa Sugiura, James W. Reinhardt, Toshiharu Shinoka.

**Data curation:** Yuichi Matsuzaki, Shinka Miyamoto, Hideki Miyachi, Chang Yu-Chun, Jacob Zbinden.

**Formal analysis:** Yuichi Matsuzaki, Shinka Miyamoto, Hideki Miyachi, Chang Yu-Chun, Jacob Zbinden.

**Funding acquisition:** Shinka Miyamoto, Christopher K. Breuer, Toshiharu Shinoka.

**Investigation:** Yuichi Matsuzaki, Shinka Miyamoto, Hideki Miyachi, Tadahisa Sugiura, Chang Yu-Chun, Jacob Zbinden.

**Methodology:** Yuichi Matsuzaki, Hideki Miyachi.

**Resources:** Christopher K. Breuer.

**Supervision:** Tadahisa Sugiura, Toshiharu Shinoka.

**Validation:** Yuichi Matsuzaki, Tadahisa Sugiura, James W. Reinhardt.

**Visualization:** Yuichi Matsuzaki, Shinka Miyamoto, James W. Reinhardt.

**Writing – original draft:** Yuichi Matsuzaki, Shinka Miyamoto, Toshiharu Shinoka.

**Writing – review & editing:** Yuichi Matsuzaki, James W. Reinhardt, Chang Yu-Chun, Jacob Zbinden, Christopher K. Breuer, Toshiharu Shinoka.

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
