## [Decision Letter · Decision Letter 0]

3 Jan 2020

PONE-D-19-33452

The Evaluation of a Tissue-Engineered Cardiac Patch Seeded with hiPS Derived Cardiac Progenitor Cells in a Rat Left Ventricular Model

PLOS ONE

Dear Dr. Shinoka,

Thank you for submitting your manuscript to PLOS ONE. After careful consideration, we feel that it has merit but does not fully meet PLOS ONE’s publication criteria as it currently stands. Therefore, we invite you to submit a revised version of the manuscript that addresses the all points raised during the review process.

We would appreciate receiving your revised manuscript by Feb 17 2020 11:59PM. To enhance the reproducibility of your results, we recommend that if applicable you deposit your laboratory protocols in protocols.io, where a protocol can be assigned its own identifier (DOI) such that it can be cited independently in the future. For instructions see: http://journals.plos.org/plosone/s/submission-guidelines#loc-laboratory-protocols

We look forward to receiving your revised manuscript.

Kind regards,

Yiru Guo, M.D., F.A.H.A.

Academic Editor

PLOS ONE

Journal Requirements:

"All animals received humane care in compliance with the National Institutes of Health (NIH) guideline for the Care and Use of Laboratory Animals. The Institutional Animal Care and Use Committee at Nationwide Children’s Hospital approved the use of animals and procedures described in this study(AR12-00079). Adult male nude athymic rats (Jackson Laboratories, Bar Harbor, ME) weighing 230–300 g were used for the left ventricle wall replacement procedure. General anesthesia was induced with ketamine (50 mg/kg, i.p) and xylazine (5 mg/kg, i.p); anesthesia was maintained using isoflurane (1.5%) in oxygen. Animals were intubated with a 16-gauge catheter and respiration was maintained at 60 cycles per minute with a tidal volume of 2.5 ml. ".

a.) Please amend your current ethics statement to confirm that your named ethics committee specifically approved this study.

For additional information about PLOS ONE submissions requirements for ethics oversight of animal work, please refer to http://journals.plos.org/plosone/s/submission-guidelines#loc-animal-research  

b.) Once you have amended this/these statement(s) in the Methods section of the manuscript, please add the same text to the “Ethics Statement” field of the submission form (via “Edit Submission”).

3. Our internal editors have evaluated your manuscript and determined that it is within the scope of our Stem Cell Plasticity in Tissue Repair and Regeneration Call for Papers. This collection of papers is headed by a team of Guest Editors for PLOS ONE and will encompass a diverse range of research articles. Additional information can be found on our announcement page: (https://collections.plos.org/s/stem-cell). If you would like your manuscript to be considered for this collection, please let us know in your cover letter and we will ensure that your paper is treated as if you were responding to this call. If you would prefer to remove your manuscript from collection consideration, please specify this in the cover letter.

[This work was supported by in part by a grant from the National Institutes of Health

(R01- HL098228 to Dr. Breuer). Dr. Shinoka receives grant support from Gunze Ltd.

(Kyoto, Japan). The is study was performed by the authors, who also had full control

of the study design, methods used, outcome measurements, data analysis, and

production of the written report.].

i) We note that you have provided funding information that is not currently declared in your Funding Statement. However, funding information should not appear in the Acknowledgments section or other areas of your manuscript. We will only publish funding information present in the Funding Statement section of the online submission form.

ii) Please remove any funding-related text from the manuscript and let us know how you would like to update your Funding Statement. Currently, your Funding Statement reads as follows:

 [Dr. Matsuzaki was the recipient of funding award from Uehara Memorial Foundation (Tokyo Japan) in 2019

This work was supported by in part by a grant from the National Institutes of Health (R01- HL098228 to Dr. Breuer). Dr. Shinoka receives grant support from Gunze Ltd. (Kyoto, Japan). The is study was performed by the authors, who also had full control of the study design, methods used, outcome measurements, data analysis, and production of the written report.

The funders had no role in study design, data collection and analysis, decision to publish, or preparation of the manuscript.].

iii) Additionally, because some of your funding information pertains to commercial funding, we ask you to provide an updated Competing Interests statement, declaring all sources of commercial funding.

iv) In your Competing Interests statement, please confirm that your commercial funding does not alter your adherence to PLOS ONE Editorial policies and criteria by including the following statement: "This does not alter our adherence to PLOS ONE policies on sharing data and materials.” as detailed online in our guide for authors  http://journals.plos.org/plosone/s/competing-interests.  If this statement is not true and your adherence to PLOS policies on sharing data and materials is altered, please explain how.

v) Please include the updated Competing Interests Statement and Funding Statement in your cover letter. We will change the online submission form on your behalf.

Reviewers' comments:

Reviewer's Responses to Questions

**Comments to the Author**

1. Is the manuscript technically sound, and do the data support the conclusions?

Reviewer #1: Yes

Reviewer #2: Partly

Reviewer #3: Partly

2. Has the statistical analysis been performed appropriately and rigorously? 

Reviewer #1: Yes

Reviewer #2: No

Reviewer #3: No

3. Have the authors made all data underlying the findings in their manuscript fully available?

Reviewer #1: Yes

Reviewer #2: Yes

Reviewer #3: Yes

4. Is the manuscript presented in an intelligible fashion and written in standard English?

Reviewer #1: Yes

Reviewer #2: Yes

Reviewer #3: Yes

5. Review Comments to the Author

Reviewer #1: Matsuzaki and colleagues submit their manuscript entitled, “The Evaluation of a Tissue-Engineered Cardiac Patch Seeded with hiPS Derived Cardiac Progenitor Cells in a Rat Left Ventricular Model” for consideration of publication in PLOS ONE. The authors use a rodent model to test whether a cell-seeded biodegradable tissue-engineered patch provides beneficial effects in LV function. The experiments are well designed with appropriate controls. Unfortunately, but not surprising cell-seeding of iPS cells did not lead to an increase in cardiomyocytes. The authors do demonstrate that a biodegradable patch can be used as a viable option for ventricular rupture.

Comments

1. The potential benefit of the biodegradable patch was the ability to cell-seed the patch. However, since the cell-seeding did not lead to increased myocardial cells, authors should clarify the benefit of their patch over non-absorbable patches.

2. Inclusion of a non-biodegradable patch as a comparison to the biodegradable patch could have been included. Do the authors have any comparison data? Especially in light that at 36 weeks, there was no difference between cell-seeded and no cell-seeding. So is a biodegradable patch add benefit?

3. Was vascularization assessed? Also, was staining performed for fibroblasts?

Reviewer #2: The work described in this manuscript addresses an important issue of designing a cell-seeded cardiac patch to improve cardiac function. Successful development of such a system could improve cardiac function in adults after ventricular rupture. The limited results in this study show that biodegradable patches implanted at the left ventricular apex with or without cardiomyocytes did not improve cardiac function. The reason for this lack of effect could not be clearly identified due to limitations of the study design.

The authors provide no assessment of cardiomyocyte function or organization prior to implantation, so it unclear whether the implanted cardiomyocytes had any contractile function at the time of implantation.

The small sample size in each group (1 and 2 month time points had 2 animals in each group and the 6 and 9 each had 3 animals in each group) precluded detection of differences between the two groups except at 8 weeks for ejection fraction. The sample size is much smaller than the authors used in their prior study, reference 11.

The histological analysis of the implants did not examine for the presence of cardiomyocytes or vascularization in the patches. Thus, the reason for the difference in ejection fraction due to cell seeding could not be evaluated.

Specific Comments

1. Figure 4. Horizontal axis label is needed for each panel.

2. Figure 4. Were the changes in LV systolic and diastolic volumes over time significant? The statistical methods used indicate that t-tests were used, but the results in Figure 4 indicate that a two-factor ANOVA (time and treatment) should be used to evaluate trends with time.

Reviewer #3: This study tested the feasibility of applying stem-cell patches with a biodegradable substrate to damaged ventricular tissue to enhance long-term remodeling. Engineered tissues for cardiac repair are evolving and in-vivo studies such as are valuable results that may help the field advance toward human use.

Although the goals of the study are modest, and methods used to quantify histological changes are limited, the results do provide useful information and show that this type of tissue engineered patch with human iPS cells and some type of biodegradable substrate can integrate with the native myocardium, albeit with limited functional outcomes at the present time.

A major limitation of the study is that the application of these patches, as stated in the introduction, can help with tissue remodeling in situations such as ventricular aneurysm, but the studies were done in a much different ventricular trauma injury model. The acute injury response to the procedure here is much different than the human diseases mentioned in the introduction and as rationale for the study. This difference is not discussed in the paper.

The methods should describe how the microscopic imaging were analyzed and what exactly was quantified, including sample sizes. Also, information on orientation of the sections, since cardiac and scar tissue, the patch and the geometry of the wall are all have direction-dependence structure. Finally, better quantification of the cell numbers and degree of calcification should be done in addition to the qualitative descriptions in the results.

Part of the study is the examine the biodegradability of the patch substrate, but this was not quantified, and only vaguely described in the paper. How do we know the that “some patch material still remains and is covered by collagen” or that it actually degrades? Some kind measure to back up these statements would be useful.

The quantitative results are likely under-powered, with small sample sizes (n=2 or 3). Have the authors done any power analysis or other statistical approaches to determine if the numerical results (echo studies) are actually statistically valid? Changes in LV function is one of the stated goals of the study, and it is hard to interpret the functional results with these sample sizes.

The conclusions paragraph can be improved. The first and third sentences are not conclusions. The second sentence is the only actual statement of a conclusion, and certainly does not summarize the main findings of the paper.

Minor:

Please include page numbers for easier referencing. Also, please use indentation or spacing for new paragraphs.

Abstract, last sentence: “differentiate into cardiomyocytes survive”, correct wording.

Intro, line 4: “cell attachment or migration” migration of what cell type?

Intro, line 14: “Litations”, Limitations??

The discussion starts with references to post-infarction remodeling, but the current study does not use that model. It would be better for the rationale for the current experiments to be better related to the same type of remodeling that would occur with the current animal model. Why is “infection” mentioned, when this is not part of the current study (but durability and biocompatibility are)?

Discussion, first para: “sustain regular rhythm”, why mention this is such a short “intro” in the discussion, when it is not part of the current study?

Discussion: “In the past, heart tissue was considered non-renewable and the heart was considered the final organ”. Wording is awkward.

Discussion: “….to create a pulsation model.” Not clear what this type of model refers to.

Discussion: “and host species leading to a non-sychronized heartbeat” not-clear that this is the case or that this would happen here. Also typo in “synchronized”

6. PLOS authors have the option to publish the peer review history of their article (what does this mean?). If published, this will include your full peer review and any attached files.

Reviewer #1: No

Reviewer #2: Yes: George A. Truskey

Reviewer #3: No

---

## [Author Response · Author response to Decision Letter 0]

7 Apr 2020

Review Comments to the Author

Reviewer #1: Matsuzaki and colleagues submit their manuscript entitled, “The Evaluation of a Tissue-Engineered Cardiac Patch Seeded with hiPS Derived Cardiac Progenitor Cells in a Rat Left Ventricular Model” for consideration of publication in PLOS ONE. The authors use a rodent model to test whether a cell-seeded biodegradable tissue-engineered patch provides beneficial effects in LV function. The experiments are well designed with appropriate controls. Unfortunately, but not surprising cell-seeding of iPS cells did not lead to an increase in cardiomyocytes. The authors do demonstrate that a biodegradable patch can be used as a viable option for ventricular rupture.

Comments

1. The potential benefit of the biodegradable patch was the ability to cell-seed the patch. However, since the cell-seeding did not lead to increased myocardial cells, authors should clarify the benefit of their patch over non-absorbable patches.

Thank you for pointing out. The discussion section has been modified. (Page 14 line 227-Page 15 line 244).Current non-biodegradable material which is used for LV repair has the risk of thrombosis and the need for frequent reoperation due to infection. Bioabsorbable materials are more useful than others as they serve as scaffolds for ingrowth of functional host myocardium and once degraded there is no longer a foreign material that can act as a nidus for infection.

2. Inclusion of a non-biodegradable patch as a comparison to the biodegradable patch could have been included. Do the authors have any comparison data? Especially in light that at 36 weeks, there was no difference between cell-seeded and no cell-seeding. So is a biodegradable patch add benefit?

Thanks for asking a question. As stated in question 1, we thought that bioabsorbable materials are useful for models of left ventricular formation after myocardial infarction. The design of this experiment focuses most on demonstrating the efficacy of iPS cell seeding using the bioabsorbable material. Therefore, the control group is only the non-seeded bioabsorbable material group. (page14 line 227-page15 line 244)

3. Was vascularization assessed? Also, was staining performed for fibroblasts?

Thank you for pointing out. We decided to evaluate angiogenesis in one-month and 6 month specimens by immunohistochemisty of CD31. It was newly confirmed by CD31 that the number of new blood vessels was higher in the seeded group. We did not stain for fibroblasts. The reason was difficulty in using immunohistochemical markers specific to markers and not other cell types capable of ECM production. (Page13 line 213-221 )

Reviewer #2: The work described in this manuscript addresses an important issue of designing a cell-seeded cardiac patch to improve cardiac function. Successful development of such a system could improve cardiac function in adults after ventricular rupture. The limited results in this study show that biodegradable patches implanted at the left ventricular apex with or without cardiomyocytes did not improve cardiac function. The reason for this lack of effect could not be clearly identified due to limitations of the study design.

The authors provide no assessment of cardiomyocyte function or organization prior to implantation, so it unclear whether the implanted cardiomyocytes had any contractile function at the time of implantation.

Cardiac progenitor cells purchased from CDI, thawed, and directly sprayed on bioabsorbable material. This time, it was Cardiac progenitor cells, not a differentiated cardiomyocyte, so it did not have a contractile function.

The small sample size in each group (1 and 2 month time points had 2 animals in each group and the 6 and 9 each had 3 animals in each group) precluded detection of differences between the two groups except at 8 weeks for ejection fraction. The sample size is much smaller than the authors used in their prior study, reference 11.

Thank you for pointing out. Small sample size is one of the limitations. This study is still at a preliminary stage, and only examines how seeding iPS cells with bioabsorbable material contributes to myocardial regeneration. In the future study, we would like to perform power analysis and increase the number of samples. (Page 18 line 298-302)

The histological analysis of the implants did not examine for the presence of cardiomyocytes or vascularization in the patches. Thus, the reason for the difference in ejection fraction due to cell seeding could not be evaluated.

Thank you for pointing out. When statistical processing was actually performed using the AVOVA, there was no significant difference in the ejection fraction. We now include in the manuscript quantification of vascularization using IHC against CD31 and found that there was a greater number of new blood vessels in the seeded group. (Figure 6) Cardiomyocytes on the patch were also described in Figure 5C. (Page 13 Line 207-221)

Specific Comments

1. Figure 4. Horizontal axis label is needed for each panel. Horizontal axis labels are now included for all panels. 

2. Figure 4. Were the changes in LV systolic and diastolic volumes over time significant? The statistical methods used indicate that t-tests were used, but the results in Figure 4 indicate that a two-factor ANOVA (time and treatment) should be used to evaluate trends with time. 

We performed statistical processing with ANOVA and there was no longer a significant difference between groups at the 2M time point. The manuscript has been updated to reflect this change. (Page 12 line 190-194)

Reviewer #3: This study tested the feasibility of applying stem-cell patches with a biodegradable substrate to damaged ventricular tissue to enhance long-term remodeling. Engineered tissues for cardiac repair are evolving and in-vivo studies such as are valuable results that may help the field advance toward human use.

Although the goals of the study are modest, and methods used to quantify histological changes are limited, the results do provide useful information and show that this type of tissue engineered patch with human iPS cells and some type of biodegradable substrate can integrate with the native myocardium, albeit with limited functional outcomes at the present time.

A major limitation of the study is that the application of these patches, as stated in the introduction, can help with tissue remodeling in situations such as ventricular aneurysm, but the studies were done in a much different ventricular trauma injury model. The acute injury response to the procedure here is much different than the human diseases mentioned in the introduction and as rationale for the study. This difference is not discussed in the paper.

Thank you for pointing out. Certainly, the indication described in the introduction is the creation of patches for use after human myocardial infarction. This study is still at a preliminary stage, and only examines how seeding iPS cells with bioabsorbable material contributes to myocardial regeneration. (Page 18 line 298-302) We believe that this model of rat is useful for evaluating remodeling for about one year. Also many paper exist to attach these iPS patch on the above of LV. But there is no report of survival experiments in rats which simulating the left ventricle, that is, simulating VSP or left ventricular rupture, and we consider the importance of this research on the method of surviving the rat for one year. (Page17 line276-279) 

1.The methods should describe how the microscopic imaging were analyzed and what exactly was quantified, including sample sizes. 

Also, 2. information on orientation of the sections, since cardiac and scar tissue, the patch and the geometry of the wall are all have direction-dependence structure. 

Finally,3. better quantification of the cell numbers and degree of calcification should be done in addition to the qualitative descriptions in the results.

Thank you for pointing out.

1. We have added details on the analysis method of the microscopy. (Page 10 line161-Page11 line172) 

2. Details have been added to the section. After the sacrifice, the left ventricle was separated from the heart by axial dissection, and a sagittal transection centered on the apical patch was performed. (Page10 line156-157)

3. Regarding the area of calcification, there was no statistically significant difference between Seeded (N = 6) 16.0 ± 5.2% vs Unseeded 8.14 ± 3.3% (N = 6) (P = 0.09).(Figure 6C-2) We stained CD31 to check for angiogenesis. There is statistically significant differences at 6M timepoint (N=3). It is defined as above the patch and near the anastomosis. (Page 13 line 207- 221) (Figure 6C-3)

Part of the study is the examine the biodegradability of the patch substrate, but this was not quantified, and only vaguely described in the paper. How do we know the that “some patch material still remains and is covered by collagen” or that it actually degrades? Some kind measure to back up these statements would be useful.

Thank you for pointing out. We characterized scaffold degradation and neotissue formation using polarized light images of Picro-Sirius Red (PSR) stained sections from Rat Patch obtained 1 M, 2M, 6M,9M after implantation, which revealed both the degradation of PGA fibers and the deposition and maturation of collagen fibers. The PGA fibers remained highly organized within the scaffold at 1M after implantation, but had thinned and begun to show evidence of early fragmentation at 2M. Only rare thin individual fragments of the PGA fibers were visible by 6M after implantation. (Page 12 line 200-204, Figure 5B) 

The quantitative results are likely under-powered, with small sample sizes (n=2 or 3). Have the authors done any power analysis or other statistical approaches to determine if the numerical results (echo studies) are actually statistically valid? Changes in LV function is one of the stated goals of the study, and it is hard to interpret the functional results with these sample sizes.

The small sample size is a limitation of this preliminary study. Power size was not verified. (page 18 line 298-302) A new statistical analysis was performed using ANOVA. (page11 line 175-178) 

The conclusions paragraph can be improved. The first and third sentences are not conclusions. The second sentence is the only actual statement of a conclusion, and certainly does not summarize the main findings of the paper.

Thank you for point out. We have modified the conclusions. (Page19 line313 -317)

Minor:

Please include page numbers for easier referencing. Also, please use indentation or spacing for new paragraphs.

Thank you for pointing out. The manuscript now includes page numbers.

Abstract, last sentence: “differentiate into cardiomyocytes survive”, correct wording.

Thank you for pointing out. 

This wording has been changed to “Seeded hiPS-CPCs did not differentiate into cardiomyocytes, but the greater number of new blood vessels in seeded patches suggests the presence of cell seeding early in the remodeling process might provide a prolonged effect on neotissue formation.” (page 4, Line 62-67)

Intro, line 4: “cell attachment or migration” migration of what cell type?

This line has been modified to specify macrophage migration. (Page 5 Line 72-74)

Intro, line 14: “Litations”, Limitations?? 

The spelling of this word has been corrected to “limitations”(page 5 Line 83) 

The discussion starts with references to post-infarction remodeling, but the current study does not use that model. It would be better for the rationale for the current experiments to be better related to the same type of remodeling that would occur with the current animal model. Why is “infection” mentioned, when this is not part of the current study (but durability and biocompatibility are)? 

Thank you for your comments and great questions to our experiments. When we try to create infarction model with the additional incision of left ventricular apex for placing the patch in rats, animal survival rates were low following the procedure. Therefore, we simplified the model to utilize left ventricular apex incision to implant our patches in the blood stream. (Page17 line 271-274)

In general, cardiac surgeons have concerns about implanting foreign materials in the heart as infective endocarditis can be a lethal complication after open heart surgery. Even now, some surgeons inevitably use Dacron patch or PTFE patch to repair for ventricular septal rupture after myocardial infarction. These clinical demands motivated us to develop new bioabsorbable materials for use in cardiac surgery. This experiment is initial attempt to develop new cell-based materials. (page 14 line227- 237)

Discussion, first para: “sustain regular rhythm”, why mention this is such a short “intro” in the discussion, when it is not part of the current study?

Thank you for pointed out. As you mentioned this is not part of the current study so we eliminated this sentence. 

Discussion: “In the past, heart tissue was considered non-renewable and the heart was considered the final organ”. Wording is awkward.

This section has been modified.　(page15 line246)

Discussion: “….to create a pulsation model.” Not clear what this type of model refers to.

This section has been modified (page16 line269)

Discussion: “and host species leading to a non-sychronized heartbeat” not-clear that this is the case or that this would happen here. Also typo in “synchronized” 

This section has been modified.　(page18 line298-page19 line310)

---

## [Decision Letter · Decision Letter 1]

24 Apr 2020

PONE-D-19-33452R1

The Evaluation of a Tissue-Engineered Cardiac Patch Seeded with hiPS Derived Cardiac Progenitor Cells in a Rat Left Ventricular Model

PLOS ONE

Dear Dr. Shinoka,

Thank you for submitting your manuscript to PLOS ONE. After careful consideration, we feel that it has merit but does not fully meet PLOS ONE’s publication criteria as it currently stands. Therefore, we invite you to submit a revised version of the manuscript that addresses the points raised during the review process.

We would appreciate receiving your revised manuscript by Jun 08 2020 11:59PM. To enhance the reproducibility of your results, we recommend that if applicable you deposit your laboratory protocols in protocols.io, where a protocol can be assigned its own identifier (DOI) such that it can be cited independently in the future. For instructions see: http://journals.plos.org/plosone/s/submission-guidelines#loc-laboratory-protocols

We look forward to receiving your revised manuscript.

Kind regards,

YIRU GUO, M.D., F.A.H.A.

Academic Editor

PLOS ONE

Reviewers' comments:

Reviewer's Responses to Questions

**Comments to the Author**

1. If the authors have adequately addressed your comments raised in a previous round of review and you feel that this manuscript is now acceptable for publication, you may indicate that here to bypass the “Comments to the Author” section, enter your conflict of interest statement in the “Confidential to Editor” section, and submit your "Accept" recommendation.

Reviewer #1: (No Response)

Reviewer #2: All comments have been addressed

Reviewer #3: All comments have been addressed

2. Is the manuscript technically sound, and do the data support the conclusions?

Reviewer #1: Yes

Reviewer #2: Partly

Reviewer #3: (No Response)

3. Has the statistical analysis been performed appropriately and rigorously? 

Reviewer #1: Yes

Reviewer #2: Yes

Reviewer #3: (No Response)

4. Have the authors made all data underlying the findings in their manuscript fully available?

Reviewer #1: Yes

Reviewer #2: Yes

Reviewer #3: (No Response)

5. Is the manuscript presented in an intelligible fashion and written in standard English?

Reviewer #1: Yes

Reviewer #2: Yes

Reviewer #3: (No Response)

6. Review Comments to the Author

Reviewer #1: Matsuzaki and colleagues submit their revised manuscript for evaluation for publication in PlosOne. The initial criticism which dampened this reviewer’s enthusiasm for the manuscript was the negative results that cell-seeding of iPS cells did not lead to an increase in cardiomyocytes nor longer term improvement of cardiac function. Additionally, the lack of an adequate control for the biodegradable patch was excluded. The authors have only made minimal changes to the manuscript in the discussion highlighting the negative results, but more importantly not providing adequate discussion of the potential positive effects of the biodegradable patch without cell seeding. There is a an abundance of literature of the use of bioactive scaffolds without the use of cell transplantation for myocardial repair/regeneration.

Reviewer #2: While preliminary, the results show that merely adding the iPS-derived cardiac progenitor cells is insufficient to ensure the survival of these cells in a cardiac patch. This information is worth transmitting to the research community. Study limitations are duly noted.

Reviewer #3: (No Response)

7. PLOS authors have the option to publish the peer review history of their article (what does this mean?). If published, this will include your full peer review and any attached files.

Reviewer #1: No

Reviewer #2: No

Reviewer #3: No

---

## [Author Response · Author response to Decision Letter 1]

28 Apr 2020

Reviewer #1: Matsuzaki and colleagues submit their revised manuscript for evaluation for publication in PlosOne. The initial criticism which dampened this reviewer’s enthusiasm for the manuscript was the negative results that cell-seeding of iPS cells did not lead to an increase in cardiomyocytes nor longer term improvement of cardiac function. Additionally, the lack of an adequate control for the biodegradable patch was excluded. The authors have only made minimal changes to the manuscript in the discussion highlighting the negative results, but more importantly not providing adequate discussion of the potential positive effects of the biodegradable patch without cell seeding. There is an abundance of literature of the use of bioactive scaffolds without the use of cell transplantation for myocardial repair/regeneration.

Thank you for your comments. As you pointed out, I have made the following changes to the discussion section, incorporating literature on the benefits of cell-free scaffolds.　Thank you for your understanding. (P14, line237- P16, line 261)

Complications associated with non-biodegradable materials and the potential to regenerate functional myocardium associated with tissue engineering have fostered the development of surgical models and evaluation of various scaffold designs. Due to challenges and limitations in the use of exogenous cell-based biomaterials, cell-free strategies are being investigated as an alternative. This approach requires the attachment and growth of host parenchymal cells as well as their production and organization of extracellular matrix. Materials of natural origin, such as collagen and fibrin, have been evaluated as scaffolds for tissue regeneration after myocardial infarction. These are applied surgically as patches, or less invasively as gel forming injectable ECM proteins. Acellular scaffolds are advantageous over cellular scaffolds in that: (1) these are off-the-shelf products that can be immediately implanted (e.g., SynerGraft®), (2) elicit a limited immune response, and (3) cost far less to produce. 14,15 Biodegradable synthetics, like the polymers that comprise our scaffold, are another category or materials widely used in tissue engineering applications due to their mechanical properties, material uniformity, stability, and lower risk of infection compared to natural biomaterials. Biodegradable synthetic polymers can be modified with high precision to meet tissue-specific properties such as appropriate degradation rates, porosity, and mechanical strength.16-19

Reviewer #2: While preliminary, the results show that merely adding the iPS-derived cardiac progenitor cells is insufficient to ensure the survival of these cells in a cardiac patch. This information is worth transmitting to the research community. Study limitations are duly noted.

Thank you for your comments. 

Reviewer #3: (No Response)

---

## [Decision Letter · Decision Letter 2]

19 May 2020

The Evaluation of a Tissue-Engineered Cardiac Patch Seeded with hiPS Derived Cardiac Progenitor Cells in a Rat Left Ventricular Model

PONE-D-19-33452R2

Dear Dr. Shinoka,

We are pleased to inform you that your manuscript has been judged scientifically suitable for publication and will be formally accepted for publication once it complies with all outstanding technical requirements.

With kind regards,

YIRU GUO, M.D., F.A.H.A.

Academic Editor

PLOS ONE

Additional Editor Comments (optional):

Reviewers' comments:

Reviewer's Responses to Questions

**Comments to the Author**

1. If the authors have adequately addressed your comments raised in a previous round of review and you feel that this manuscript is now acceptable for publication, you may indicate that here to bypass the “Comments to the Author” section, enter your conflict of interest statement in the “Confidential to Editor” section, and submit your "Accept" recommendation.

Reviewer #1: All comments have been addressed

2. Is the manuscript technically sound, and do the data support the conclusions?

Reviewer #1: Yes

3. Has the statistical analysis been performed appropriately and rigorously? 

Reviewer #1: Yes

4. Have the authors made all data underlying the findings in their manuscript fully available?

Reviewer #1: Yes

5. Is the manuscript presented in an intelligible fashion and written in standard English?

Reviewer #1: Yes

6. Review Comments to the Author

Reviewer #1: Authors addressed concerns. No further comments or revisions needed. Although a negative study, the manuscript further provides evidence that iPS cells alone are not beneficial for myocardial repair and that a polymer alone may impact LV remodeling after an injury.

7. PLOS authors have the option to publish the peer review history of their article (what does this mean?). If published, this will include your full peer review and any attached files.

Reviewer #1: No

---

## [Editor Report · Acceptance letter]

28 May 2020

PONE-D-19-33452R2 

The Evaluation of a Tissue-Engineered Cardiac Patch Seeded with hiPS Derived Cardiac Progenitor Cells in a Rat Left Ventricular Model 

Dear Dr. Shinoka:

I am pleased to inform you that your manuscript has been deemed suitable for publication in PLOS ONE. Congratulations! Your manuscript is now with our production department. 

With kind regards,

on behalf of

Dr. YIRU GUO 

Academic Editor

PLOS ONE